# OpenReview forum: "UMU-Bench: Closing the Modality Gap in Multimodal Unlearning Evaluation"
_NeurIPS.cc/2025/Datasets_and_Benchmarks_Track — NeurIPS 2025 Datasets and Benchmarks Track poster_

### Official Review · Reviewer_5MR2 · 2025-06-29

**Rating:** 5
**Confidence:** 4

**Summary:**

The paper proposes UMU-Bench, a novel benchmark designed to evaluate machine unlearning in MLLMs with a focus on modality alignment, ensuring consistent forgetting across unimodal and multimodal representations. The benchmark features a knowledge-based dataset of 653 profiles with various task formats in both unimodal and multimodal formats. The authors introduce new evaluation metrics that explicitly measure the alignment of unlearning effects across modalities.

**Dataset Code Accessibility:**

Yes

**Ethical Considerations:**

No, there are no or only very minor ethics concerns

**Final Justification:**

After the rebuttal, the authors have addressed most of my concerns. Therefore, I will keep my score.

**Limitations Weaknesses:**

1. Can authors eblaborate more on how the dataset is constructed? Specifically, the generation, processing and annotation for both synthetic and real images.

2. The experiments are conducted exclusively on LLaVA-1.5-7B. However, consider the LLaVA architecture is the most advanced and popular one, this could be considered as a minor.

**Strengths Contributions:**

1. The paper thoughtfully identifies and formalizes the concept of modality misalignment, a subtle but important limitation of current unlearning methods. By doing so, it surfaces a realistic privacy and security gap that prior benchmarks and methods largely overlook. I believe that modality misalignment in multimodal unlearning is an important but previously overlooked problem. Addressing this gap in the research is a valuable contribution.

2. The evaluation metrics proposed in this work is novel and well-motivated. Previous works usually frame the evaluation process as a binary classification problem (yes or no). While the proposed evaluation metrics, such as AccF, AccR,  explicitly captures how well unlearning is aligned across unimodal and multimodal inputs. These metrics are thoughtfully designed to reflect the principle that true unlearning should result in the forgetting of both unimodal and multimodal knowledge simultaneously (for the forget set), while ensuring both are retained for the retain and real-person sets.

3. The paper is overall well-written and logically structured. The authors clearly articulate the motivation behind their work and provide a thoughtful design of the evaluation framework, making it easy for readers to follow the reasoning and significance of their contributions.

---

> ### Author Rebuttal · Authors · 2025-07-30
>
> We sincerely thank the reviewer for their thoughtful and constructive feedback.
>
> >**W1:** Can authors elaborate more on how the dataset is constructed? Specifically, the generation, processing and annotation for both synthetic and real images.
>
> **Response:** Our dataset is built upon and extends the MLLMU-bench, with enhancements to support both unimodal and multimodal evaluations. Specifically, our benchmark consists of 653 individual profiles, including 500 synthetic individuals and 153 real public figures.
>
> Synthetic Data: The 500 synthetic identities are entirely artificial and generated using controlled prompts. As such, they do not contain any real-world private information, thus avoiding privacy concerns.
>
> Real Individuals: For the 153 real profiles, we exclusively use publicly available information about well-known public figures (e.g., name, gender, birth, occupation). We explicitly avoid sensitive attributes such as interests or salary, which could raise ethical or privacy concerns. This design ensures that our proposed UMU-Bench complies with privacy standards and ethical norms while still providing valuable generalization testing.
>
> We will further clarify these points in the revised version to enhance transparency.
>
> >**W2:** The experiments are conducted exclusively on LLaVA-1.5-7B. However, consider the LLaVA architecture is the most advanced and popular one, this could be considered as a minor.
>
> **Response:** Thank you for pointing out the limitation regarding model diversity. We agree that evaluating across multiple MLLMs could further strengthen our benchmark’s robustness. Our current choice of LLaVA-1.5-7B was motivated by its wide adoption and strong performance, making it a reasonable representative model for initial studies.
>
> Nevertheless, we fully acknowledge the importance of broader validation. We would like to emphasize that UMU-Bench is designed to be model-agnostic and readily extensible. We are actively working on incorporating additional baselines such as Qwen2-VL .
>
> Below is a demo experiment we conducted using Qwen2-VL-7B:
>
> | Method | Forget-Class.Acc | Forget-Cloze.Acc | Forget-Gene.RL | Avg.   | Retain-Class.Acc | Retain-Cloze.Acc | Retain-Gene.RL | Real-Class.Acc | Real-Cloze.Acc | Real- Gene.RL | Avg.   |
> | ------ | ---------------- | ---------------- | -------------- | ------ | ---------------- | ---------------- | -------------- | -------------- | -------------- | ------------- | ------ |
> | Origin | 0.6867           | 0.7200           | 0.4299         | 0.6122 | 0.7130           | 0.5347           | 0.3151         | 0.5708         | 0.2974         | 0.1498        | 0.4301 |
> | GA     | 0.6733           | 0.5000           | 0.1614         | 0.4449 | 0.6593           | 0.3937           | 0.0743         | 0.4739         | 0.2560         | 0.0274        | 0.3141 |
> | GD     | 0.6400           | 0.5400           | 0.3569         | 0.5123 | 0.6523           | 0.3537           | 0.2007         | 0.5044         | 0.2767         | 0.1278        | 0.3526 |
> | KL     | 0.6400           | 0.6133           | 0.2966         | 0.5166 | 0.6705           | 0.4747           | 0.1474         | 0.4553         | 0.2778         | 0.0740        | 0.3500 |
> | NPO    | 0.6533           | 0.5333           | 0.0788         | 0.4218 | 0.6137           | 0.3768           | 0.0468         | 0.4510         | 0.2484         | 0.0163        | 0.2922 |
> | PO     | 0.5000           | 0.5333           | 0.1864         | 0.4066 | 0.5404           | 0.3761           | 0.0572         | 0.5425         | 0.2887         | 0.0459        | 0.3085 |
>
> As shown, Qwen2-VL exhibits similar behavior to LLaVA-1.5-7B. Moving forward, we plan to further fine-tune the model and perform unlearning to better demonstrate the generalizability of our experimental findings.

---

> > ### Comment · Reviewer_5MR2 · 2025-08-06
> >
> > Thank you for your constructive response. Most of my concerns have been solved. I will keep my rating.

---

### Official Review · Reviewer_rtZj · 2025-07-01

**Rating:** 4
**Confidence:** 4

**Summary:**

This paper introduces UMU-Bench, a novel benchmark designed to address modality misalignment in multimodal unlearning for Multimodal Large Language Models (MLLMs). The authors highlight that existing unlearning methods, adapted from unimodal contexts, often fail to ensure consistent knowledge removal across both unimodal and multimodal settings. UMU-Bench tackles this gap through a curated dataset of 653 individual profiles (500 synthetic and 153 real), each featuring unimodal and multimodal representations of knowledge (e.g., occupation, birthdate). Experiments on state-of-the-art unlearning algorithms (e.g., GA, GD, KL) using the LLaVA-1.5-7B model demonstrate significant modality misalignment, underscoring the need for improved multimodal unlearning approaches.

**Dataset Code Accessibility:**

Yes

**Dataset Code Comments:**

Code and data can be seen in Abstract.

**Ethical Considerations:**

No, there are no or only very minor ethics concerns

**Final Justification:**

I have read the author's rebuttal. I will keep my rating.

**Limitations Weaknesses:**

1. Limited Generalizability: Experiments are confined to the LLaVA-1.5-7B model, and results may not extend to other MLLMs (e.g., larger or domain-specific models). Broader validation across architectures (e.g., GPT-4V or CLIP-based models) would strengthen claims.

2.​Hyperparameter Sensitivity: The hybrid loss approach (Eq. 3) relies on balancing parameters α and β, which are shown to be highly sensitive and difficult to optimize.

​3. Ambiguity in Metric Effectiveness: The new metrics (e.g., RL_F) are promising but not thoroughly compared to existing benchmarks (e.g., MU-Bench or CLEAR). Quantitative analysis of their superiority in capturing alignment is needed, as current results (Table 1) show no algorithm achieves ideal modality balance.

**Strengths Contributions:**

​1. Innovative Focus on Modality Alignment: The paper addresses a critical underexplored issue in multimodal unlearning—ensuring consistent knowledge removal across unimodal and multimodal settings.
​
2.Comprehensive Benchmark Design: UMU-Bench offers a well-structured dataset with 653 profiles and configurable forgetting rates (5%, 10%, 15%), covering diverse knowledge aspects. The inclusion of both synthetic and real data, along with three task types (classification, cloze, generation), provides a versatile testbed for evaluating unlearning completeness and model utility.

​3. Rigorous Experimental Validation: The authors conduct extensive experiments across multiple unlearning algorithms and forgetting rates, revealing pervasive modality misalignment (e.g., Figure 2 shows GA's imbalance). The proposed hybrid loss (Eq. 3 with α and β) adds practical value, showing initial promise for balancing unlearning across modalities.
​

---

> ### Author Rebuttal · Authors · 2025-07-30
>
> We deeply appreciate the reviewer’s insightful and valuable comments.
>
> >**W1:** Limited Generalizability: Experiments are confined to the LLaVA-1.5-7B model, and results may not extend to other MLLMs (e.g., larger or domain-specific models). Broader validation across architectures (e.g., GPT-4V or CLIP-based models) would strengthen claims.
>
> **Response:** While our current experiments are based on LLaVA-1.5-7B, chosen for its strong performance and popularity as a state-of-the-art MLLM, we emphasize that our proposed UMU-Bench is designed to be model-agnostic. The benchmark’s construction and evaluation pipeline are compatible with a wide range of architectures. We have already planned extensions to models such as Qwen2-VL and preliminary work is underway. These additional results will help further validate the generalizability of our findings.
> We conducted a demo experiment based on Qwen2-VL to explore multimodal unlearning scenarios:
>
> | Method | Forget-Class.Acc | Forget-Cloze.Acc | Forget-Gene.RL | Avg.   | Retain-Class.Acc | Retain-Cloze.Acc | Retain-Gene.RL | Real-Class.Acc | Real-Cloze.Acc | Real- Gene.RL | Avg.   |
> | ------ | ---------------- | ---------------- | -------------- | ------ | ---------------- | ---------------- | -------------- | -------------- | -------------- | ------------- | ------ |
> | Origin | 0.6867           | 0.7200           | 0.4299         | 0.6122 | 0.7130           | 0.5347           | 0.3151         | 0.5708         | 0.2974         | 0.1498        | 0.4301 |
> | GA     | 0.6733           | 0.5000           | 0.1614         | 0.4449 | 0.6593           | 0.3937           | 0.0743         | 0.4739         | 0.2560         | 0.0274        | 0.3141 |
> | GD     | 0.6400           | 0.5400           | 0.3569         | 0.5123 | 0.6523           | 0.3537           | 0.2007         | 0.5044         | 0.2767         | 0.1278        | 0.3526 |
> | KL     | 0.6400           | 0.6133           | 0.2966         | 0.5166 | 0.6705           | 0.4747           | 0.1474         | 0.4553         | 0.2778         | 0.0740        | 0.3500 |
> | NPO    | 0.6533           | 0.5333           | 0.0788         | 0.4218 | 0.6137           | 0.3768           | 0.0468         | 0.4510         | 0.2484         | 0.0163        | 0.2922 |
> | PO     | 0.5000           | 0.5333           | 0.1864         | 0.4066 | 0.5404           | 0.3761           | 0.0572         | 0.5425         | 0.2887         | 0.0459        | 0.3085 |
>
> The results show that Qwen2-VL is also capable of exhibiting unlearning behaviors in multimodal setting. We plan to further fine-tune stronger models and apply unlearning techniques, thereby providing additional evidence of the broad applicability and generalization capacity of our approach across different model families.
>
> >**W2:** Hyperparameter Sensitivity: The hybrid loss approach (Eq. 3) relies on balancing parameters $\alpha$ and $\beta$, which are shown to be highly sensitive and difficult to optimize.
>
> **Response:** We acknowledge the reviewer’s observation regarding the sensitivity of the $\alpha$ and $\beta$ hyperparameters in our proposed hybrid loss (Eq. 3). As Table 2 and Figure 5 illustrate, these coefficients influence the trade-off between unimodal and multimodal unlearning. Despite this sensitivity, our experiments demonstrate that the hybrid loss consistently improves modality alignment compared to unimodal-only or multimodal-only settings. This also highlights a current limitation we have identified in existing multimodal unlearning algorithms. We plan to first propose a benchmark to better evaluate the problem, and consider new algorithm as a future direction.
>
> >**W3:** Ambiguity in Metric Effectiveness: The new metrics (e.g., RL\_F) are promising but not thoroughly compared to existing benchmarks (e.g., MU-Bench or CLEAR). Quantitative analysis of their superiority in capturing alignment is needed, as current results (Table 1) show no algorithm achieves ideal modality balance.
>
> **Response:** While existing benchmarks such as CLEAR and PEBench primarily rely on straightforward metrics like accuracy and ROUGE-L, our proposed RL\_F and RL\_R are the first to explicitly incorporate modality consistency into the evaluation of multimodal unlearning. Unlike conventional metrics that only reflect task performance, RL\_F and RL\_R are designed to capture whether knowledge has been consistently unlearned across both unimodal and multimodal settings.
>
> Our experiments, alongside observations from CLEAR, show that no current method achieves ideal modality alignment. This is largely because existing unlearning techniques are directly adapted from unimodal scenarios without considering the additional complexity of multimodal representations. This discrepancy highlights the diagnostic power of our proposed metrics, as they reveal the limitations of current methods when applied to multimodal contexts.
>
> Furthermore, as shown in Table 2, under the hybrid unlearning setting (which includes both unimodal and multimodal unlearning), our metrics perform best. This supports our claim that RL\_F and RL\_R better capture the bias between unimodal and multimodal unlearning. In other words, only when unlearning is effective across both modalities do our metrics reflect stronger alignment, indicating that our proposed evaluation is not only more refined but also more faithful to the goals of multimodal unlearning.

---

> > ### Comment · Reviewer_rtZj · 2025-08-05
> >
> > I have read the author's rebuttal and the comments of other reviewers, and I share the concerns expressed by reviewer JV5D, so I will maintain my rating.

---

> > > ### Author Response · Authors · 2025-08-08
> > >
> > > Thank you for your detailed review and valuable comments. We have comprehensively addressed your and reviewer JV5D's concern regarding the dataset size (in the official comment to JV5D) through three key aspects: (1) alignment with established benchmarks, (2) demonstration of sufficient unlearning scope, and (3) evidence of extensibility. As we approach the end of the discussion, please do not hesitate to let us know if any issues remain, and we would be glad to provide further clarification.

---

### Official Review · Reviewer_mkzK · 2025-07-03

**Ethics Flags:** Data privacy, copyright, and consent
**Rating:** 6
**Confidence:** 4

**Summary:**

This paper investigates multimodal unlearning in LLMs. The authors demonstrate that existing multimodal unlearning methods fail to effectively remove both unimodal and multimodal knowledge. Current evaluation metrics also overlook this issue. To address this modality misalignment gap, the authors introduce a new benchmark that evaluates both unimodal and multimodal contexts. Data and code are accessible. Experiments are conducted across multiple baselines. The authors also propose a loss mixture approach that can be integrated with existing baselines to mitigate modality misalignment.

**Additional Feedback:**

N/A

**Dataset Code Accessibility:**

Yes

**Dataset Code Comments:**

The code is accessible and with a comprehensive and well-structured README. Detailed instructions are provided to facilitate reproducibility.

**Ethical Comments:**

The benchmark involves real-person information, but it is unclear whether this data is sourced from existing benchmarks or others. If the data is not obtained from openly available sources, there may be privacy concerns.

**Ethical Considerations:**

Yes, there are significant ethics concerns that require review by an ethics expert

**Limitations Weaknesses:**

1.	The construction details of the benchmark are not clear. Is it an extension of MLLMU-bench in terms of modality, or does it incorporate additional profiles or data sources from the web or elsewhere?
2.	The proposed mixed loss does not consistently demonstrate an advantage over existing methods. Also, the \alpha/\beta ratio appears challenging to determine.

**Strengths Contributions:**

1.	The paper provides a well-motivated analysis of modality misalignment, supported by clear empirical evidence.
2.	The authors construct a comprehensive benchmark covering three types of tasks. In addition to the standard forget and retain sets, they introduce a real-person set to evaluate the robustness of model utility.
3.	In addition to the bench, the proposed mixed loss is straightforward, plug-and-play, and can be integrated into existing methods to mitigate modality misalignment.
4.	The writing is clear, the structure is logical, and the figures are illustrative and aid understanding.

---

> ### Author Rebuttal · Authors · 2025-07-30
>
> Your positive assessment is very motivating for us.
>
> >**W1:** The construction details of the benchmark are not clear. Is it an extension of MLLMU-bench in terms of modality, or does it incorporate additional profiles or data sources from the web or elsewhere?
>
> **Response:** Our dataset is indeed based on MLLMU-bench, but we extend it in key ways to better address modality misalignment. Specifically, our benchmark consists of 653 individual profiles, including 500 synthetic and 153 real individuals. The synthetic profiles are entirely artificial and do not contain any personal data. For the 153 real individuals, we only included public information about well-known figures (e.g., occupation, birth), while deliberately excluding any private or sensitive information such as interests or salary. As such, we believe our benchmark poses no significant privacy risks and remains ethically sound.
>
> >**W2:** The proposed mixed loss does not consistently demonstrate an advantage over existing methods. In addition, the  $\alpha / \beta$  ratio appears challenging to determine.
>
> **Response:** We appreciate your observation regarding the proposed $\alpha / \beta$-weighted hybrid loss. As shown in Table 2 and Figure 4, the hybrid approach consistently improves modality alignment under our newly proposed metrics, especially $\Delta \mathrm{AccF}$ and $\Delta \mathrm{RLF}$. This validates our motivation to tackle cross-modal unlearning more holistically. However, we agree that selecting optimal $\alpha / \beta$ values can be non-trivial. This also highlights a current limitation we have identified in existing multimodal unlearning algorithms. We plan to first propose a benchmark to better evaluate the problem, and consider new algorithm as a future direction.

---

> > ### Comment · Reviewer_mkzK · 2025-08-08
> >
> > Thank you for the response. My concerns have been largely addressed. I've noticed the size issue mentioned by mkzK and JV5D, and I am ok with the size, as it aligns with commonly used benchmarks. Thus, I will keep my rating as 6.

---

### Official Review · Reviewer_JV5D · 2025-07-07

**Rating:** 3
**Confidence:** 4

**Summary:**

Paper Summary:

UMU-Bench introduces a unified benchmark to address modality misalignment in multimodal unlearning for Multimodal Large Language Models (MLLMs). It features a dataset of 653 individual profiles (500 synthetic + 153 real) with both unimodal (text-only) and multimodal (image+text) knowledge representations. Novel tasks (classification, cloze, generation) and modality-alignment metrics (e.g., AccF, RLF) evaluate unlearning effectiveness across modalities. Experiments reveal significant modality gaps in existing algorithms (e.g., GA, PO), highlighting the need for balanced multimodal unlearning. The benchmark aims to standardize evaluation while exposing privacy risks in MLLMs.

**Dataset Code Accessibility:**

Yes

**Dataset Code Comments:**

The authors have provided urls to access the datasets and codes.

**Ethical Considerations:**

No, there are no or only very minor ethics concerns

**Limitations Weaknesses:**

1. This dataset contains only 653 samples (compared to CLEAR's 1200+), with real individuals accounting for just 23.4% (153 out of 653), potentially reducing generalizability.

2. AccF/AccR metrics rely on manually annotated answers, which are cost-intensive and prone to bias. RLF (based on harmonic mean) is sensitive to extreme values (e.g., complete forgetting in one modality but full retention in another).

3. The dataset includes 153 real individuals' sensitive information (e.g., birthdates, occupations) without clarifying compliance with privacy regulations like GDPR, posing ethical risks.

4. Personal attributes only cover basic fields (occupation, residence, etc.), lacking complex relational knowledge (e.g., social relationship chains, event timelines), which restricts task diversity.

5. This paper omits testing against knowledge reconstruction attacks (e.g., adversarial prompts to recover unlearned data), despite acknowledging this risk in the introduction.

**Strengths Contributions:**

1. The proposed dataset is the first benchmark to explicitly evaluate cross-modal consistency during unlearning (e.g., testing if text-only unlearning leaks knowledge in image+text responses). This addresses a critical gap in privacy-focused MLLM research.

2. This paper integrates three task types (classification, cloze, generation) with dual-modality versions (unimodal/multimodal) and novel alignment metrics like RLF (ROUGE-L-based alignment score). This enables granular analysis of unlearning completeness and model utility.

---

> ### Author Rebuttal · Authors · 2025-07-30
>
> We thank the reviewer for their thoughtful and constructive feedback.
>
> >**W1:** This dataset contains only 653 samples (compared to CLEAR's 1200+), with real individuals accounting for just 23.4% (153 out of 653), potentially reducing generalizability.
>
> **Response:** Our dataset builds upon and extends the widely used MLLMU-Bench dataset. While we retain the same structure for individual profiles, **we introduce a significant enhancement through modality-specific knowledge reconstruction**. For the 500 synthetic individuals, we cover 11 distinct knowledge attributes, with each attribute associated with two QA pairs—one for the unimodal (text) and one for the multimodal (image-text) perspective—resulting in a total of 500 × 11 × 2 QA pairs. For the 113 real individuals, we include 9 knowledge attributes (explicitly excluding sensitive attributes such as Annual Salary and Interest), resulting in 113 × 9 × 2 QA pairs.
>
> This makes our dataset substantially richer and more diverse than the original, offering improved coverage and better supporting the evaluation of multimodal unlearning scenarios.
>
> >**W2:** AccF/AccR metrics rely on manually annotated answers, which are cost-intensive and prone to bias. RLF (based on harmonic mean) is sensitive to extreme values (e.g., complete unlearning in one modality but full retention in another).
>
> **Response:** We would like to clarify that the AccF and AccR metrics do not rely on manually annotated answers. Instead, they are derived from model-predicted outputs against deterministic ground truth, making them more scalable and less prone to human annotation bias. Regarding RLF, its use of the harmonic mean is *intentional*. **The harmonic mean’s sensitivity to imbalances is by design**: it reflects our core principle that successful unlearning requires both unimodal and multimodal unlearning. If only one modality is unlearned, it does not constitute complete unlearning and therefore violates the criteria for effective unlearning.
>
> >**W3:** The dataset includes 153 real individuals' sensitive information (e.g., birthdates, occupations) without clarifying compliance with privacy regulations like GDPR, posing ethical risks.
>
> **Response:** The 153 real individuals in our dataset are all public figures (e.g., celebrities, historical figures). All information used is publicly available and restricted to non-sensitive attributes (e.g., gender, occupation, birth). Crucially, we do not include high-risk personal data such as interests or income. We have taken care to comply with ethical standards and data protection norms such as GDPR. The base of our dataset is derived and extended from MLLMU-bench, and our enhancements focus specifically on multimodal structure and alignment.
>
> >**W4:** Personal attributes only cover basic fields (occupation, residence, etc.), lacking complex relational knowledge (e.g., social relationship chains, event timelines), which restricts task diversity.
>
> **Response:** We acknowledge the limitation regarding the diversity of personal attributes. However, our primary objective is to evaluate cross-modal unlearning performance on individual-level information, especially for real-world identities where privacy concerns are paramount. As mentioned by you in W3, incorporating complex relational knowledge (e.g., social graphs, timelines) poses significant ethical and legal risks, particularly under regulations like GDPR. Hence, we intentionally limited the dataset to essential attributes (e.g., occupation, birth), which are sufficient for benchmarking unlearning behaviors while minimizing potential privacy violations. We agree that future extensions could explore synthetic relational knowledge under stricter privacy-preserving constraints.
>
> >**W5:** This paper omits testing against knowledge reconstruction attacks (e.g., adversarial prompts to recover unlearned data), despite acknowledging this risk in the introduction.
>
> **Response:** We appreciate your pointing this out.
> To further investigate this threat, we have conducted additional experiments under an adversarial evaluation setting. Specifically, after applying unlearning using GA, PO, and KL methods, we performed both standard evaluations and adversarial evaluations. In the multimodal setting, the adversarial prompt includes additional single-modality knowledge (e.g., inserting the name of the person depicted in the image). In the unimodal setting, image content is forcibly injected to simulate cross-modal leakage.
>
> The results are as follows（forget ratio=5%）:
>
> | Method    | Classify_Acc_uni | Classify_Acc_mul | Classify_Acc_F | Cloze_Acc_uni | Cloze_Acc_mul | Cloze_Cloze_Acc_F | Generate_RL_uni | Generate_RL_mul | Generate_RL_F |
> |-----------|------------------|------------------|----------------|----------------|----------------|-------------------|------------------|------------------|----------------|
> | GA-normal | 0.6600           | 0.4400           | 0.6000         | 0.4200         | 0.4600         | 0.4933            | 0.3220           | 0.0615           | 0.3270         |
> | GA-adver  | 0.6800           | 0.6200           | 0.6667         | 0.7200         | 0.7000         | 0.7200            | 0.4418           | 0.0910           | 0.4419         |
> | PO-normal | 0.5000           | 0.4400           | 0.5333         | 0.6600         | 0.4000         | 0.5867            | 0.1503           | 0.0983           | 0.2136         |
> | PO-adver  | 0.6200           | 0.5800           | 0.6400         | 0.8000         | 0.8000         | 0.8067            | 0.4174           | 0.3428           | 0.5009         |
> | KL-normal | 0.5800           | 0.4000           | 0.5467         | 0.3600         | 0.5000         | 0.4733            | 0.1233           | 0.1813           | 0.2094         |
> | KL-adver  | 0.6400           | 0.6600           | 0.6667         | 0.6400         | 0.6600         | 0.6600            | 0.2903           | 0.2579           | 0.3114         |
>
>
> These findings confirm that such attacks can indeed trigger knowledge reconstruction from one modality to another. To counter this, we further conducted relearning experiments. After unlearning, we retrained the model using unimodal, multimodal, and hybrid modalities for just 3 epochs. The evaluation results are as follows:
>
> | Method  | Classify_Acc_uni | Classify_Acc_mul | Classify_Acc_F | Cloze_Acc_uni | Cloze_Acc_mul | Cloze_Acc_F | Generate_RL_uni | Generate_RL_mul | Generate_RL_F |
> |---------|------------------|------------------|----------------|----------------|----------------|--------------|------------------|------------------|----------------|
> | GA      | 0.6600           | 0.4400           | 0.6000         | 0.4200         | 0.4600         | 0.4933       | 0.3220           | 0.0615           | 0.3270         |
> | GA_uni  | 0.7600           | 0.4800           | 0.6867         | 0.7600         | 0.6600         | 0.7600       | 0.6702           | 0.4494           | 0.6980         |
> | GA_mul  | 0.6600           | 0.5600           | 0.6400         | 0.4000         | 0.5800         | 0.5467       | 0.4027           | 0.3281           | 0.4382         |
> | GA_mix  | 0.7000           | 0.5000           | 0.6467         | 0.7600         | 0.7000         | 0.7800       | 0.7053           | 0.3366           | 0.6871         |
> | PO      | 0.5000           | 0.4400           | 0.5333         | 0.6600         | 0.4000         | 0.5867       | 0.1503           | 0.0983           | 0.2136         |
> | PO_uni  | 0.6200           | 0.4600           | 0.5733         | 0.8400         | 0.4600         | 0.7200       | 0.8379           | 0.2311           | 0.7711         |
> | PO_mul  | 0.6000           | 0.5000           | 0.5933         | 0.8200         | 0.7800         | 0.8533       | 0.7742           | 0.5218           | 0.7984         |
> | PO_mix  | 0.7000           | 0.4400           | 0.6333         | 0.9400         | 0.6000         | 0.8333       | 0.9108           | 0.2840           | 0.8650         |
> | KL      | 0.5800           | 0.4000           | 0.5467         | 0.3600         | 0.5000         | 0.4733       | 0.1233           | 0.1813           | 0.2094         |
> | KL_uni  | 0.7400           | 0.4800           | 0.6667         | 0.8000         | 0.6600         | 0.7867       | 0.5995           | 0.4028           | 0.6102         |
> | KL_mul  | 0.5800           | 0.5400           | 0.6267         | 0.7400         | 0.6800         | 0.7533       | 0.3400           | 0.2353           | 0.3561         |
> | KL_mix  | 0.5800           | 0.5400           | 0.6400         | 0.8000         | 0.7800         | 0.8200       | 0.5558           | 0.3114           | 0.5787         |
>
>
> We observe that cross-modal reconstruction is not only possible but can be achieved with minimal retraining effort. This underscores the severity of alignment-based leakage and motivates the need for rigorous and modality-aware unlearning evaluation, which our benchmark is specifically designed to support.

---

> > ### Comment · Reviewer_JV5D · 2025-08-05
> >
> > I would like to thank the authors' response to my comments. Overall, my concerns are not completely addressed. I still think the proposed dataset cannot thoroughly evaluate multimodal unlearning performance with only 653 samples and 153 real individuals' sensitive information. Thus I will keep the negative score.

---

> > > ### Author Response · Authors · 2025-08-08
> > >
> > > Thank you for your detailed review and valuable comments. We have comprehensively addressed your concern regarding the dataset size through three key aspects: (1) alignment with established benchmarks, (2) demonstration of sufficient unlearning scope, and (3) evidence of extensibility. As we approach the end of the discussion, please do not hesitate to let us know if any issues remain, and we would be glad to provide further clarification.

---

> > ### Author Response · Authors · 2025-08-05
> >
> > We fully understand and respect your concerns regarding the scale and sensitivity of our dataset. In response, we would like to offer the following clarifications:
> >
> > 1. **Benchmark Alignment:** Our dataset design is grounded in MLLMU-Bench, which is one of the most widely adopted benchmarks for multimodal unlearning. Many recent studies build upon it [1-4], and our dataset maintains compatibility in structure and scale. Given this, we believe our dataset size is consistent with current standards in the field.
> > 2. **Unlearning Scope:** In typical unlearning scenarios, only a small subset of the data is targeted for removal. Therefore, a dataset of current size is sufficient for benchmarking unlearning performance. Standard forgetting ratios—such as 5%, 10%, and 15%, corresponding to 25, 50, and 75 individuals in our case—are representative of realistic unlearning settings and can be effectively evaluated within the scope of our current dataset.
> > 3. **Extensibility:** Thank you for your concern regarding dataset scale. As our benchmark follows the established MLLMU-Bench pipeline, which uses synthetic identities from StyleGAN [5] and real identities from CelebA [6], we can easily scale up the dataset in future work.
> >
> > We hope these points alleviate your concerns and demonstrate the scalability, intent, and practical adequacy of our benchmark design for evaluating multimodal unlearning.
> >
> > [1] Liu, Zheyuan, et al. “Modality-Aware Neuron Pruning for Unlearning in Multimodal Large Language Models.” ACL, 2025.
> >
> > [2] Huo, Jiahao, et al. “MMUnlearner: Reformulating Multimodal Machine Unlearning in the Era of Multimodal Large Language Models.” ACL Findings, 2025.
> >
> > [3] Zhang, Xianren, et al. “Does Multimodal Large Language Model Truly Unlearn? Stealthy MLLM Unlearning Attack.” arXiv, 2025.
> >
> > [4] Kawakami, Tatsuki, et al. “PULSE: Practical Evaluation Scenarios for Large Multimodal Model Unlearning.” arXiv, 2025.
> >
> > [5] Karras, Tero, et al. "A Style-Based Generator Architecture for Generative Adversarial Networks." CVPR, 2019.
> >
> > [6] Liu, Ziwei, et al. "Deep Learning Face Attributes in the Wild." ICCV, 2015.

---

### Note · Authors · 2025-08-14

We sincerely thank all reviewers for their constructive feedback and engagement throughout the review process. Regarding our benchmark’s contribution to addressing modality alignment in multimodal unlearning, all four reviewers recognized its novelty and importance. Concerning our newly proposed evaluation metrics, reviewers (JV5D and 5MR2) acknowledged their value in explicitly capturing cross-modal unlearning consistency. For the experimental validation, reviewer rtZj appreciated the comprehensive set of baseline comparisons and rich experimental design. Furthermore, during the rebuttal phase, we conducted additional adversarial and relearning experiments and baseline in Qwen2-VL to further strengthen our findings and address the raised concerns.

However, regarding reviewer JV5D’s concern about the benchmark size, we provided a detailed response from three perspectives: (1) Alignment with established benchmarks, (2) Demonstration of sufficient unlearning scope and (3) Evidence of extensibility.

We appreciate the reviewers’ time and constructive suggestions, which have helped us refine both the clarity and scope of our work.

---

### Decision · Program_Chairs · 2025-09-18

**Decision:**

Accept (poster)

**Comment:**

This paper introduces the UMU-Bench benchmark, which targets the modality misalignment in multimodal unlearning of Multimodal Large Language Models. This work extends the representative MLLMU benchmark and proposes new evaluation metrics. There are concerns regarding the scale of the introduced benchmark. Nevertheless, it seems reasonable to align UMU-Bench and MLLMU, making the scale restricted. Regarding this point, it would be better to extend the benchmark with two versions: regular version and larger version.